# CTIBench: A Benchmark for Evaluating LLMs in Cyber Threat Intelligence

**Md Tanvirul Alam**
Rochester Institute of Technology
Rochester, NY, USA

**Dipkamal Bhusal**
Rochester Institute of Technology
Rochester, NY, USA

**Le Nguyen**
Rochester Institute of Technology
Rochester, NY, USA

**Nidhi Rastogi**
Rochester Institute of Technology
Rochester, NY, USA

## Abstract

Cyber threat intelligence (CTI) is crucial in today's cybersecurity landscape, providing essential insights to understand and mitigate the ever-evolving cyber threats. The recent rise of Large Language Models (LLMs) have shown potential in this domain, but concerns about their reliability, accuracy, and hallucinations persist. While existing benchmarks provide general evaluations of LLMs, there are no benchmarks that address the practical and applied aspects of CTI-specific tasks. To bridge this gap, we introduce CTIBench, a benchmark designed to assess LLMs' performance in CTI applications. CTIBench includes multiple datasets focused on evaluating knowledge acquired by LLMs in the cyber-threat landscape. Our evaluation of several state-of-the-art models on these tasks provides insights into their strengths and weaknesses in CTI contexts, contributing to a better understanding of LLM capabilities in CTI. Code and dataset available at
`https://github.com/aiforsec/cti-bench`.

## 1   Introduction

The evolving landscape of the digital world has led to an unprecedented growth in cyber attacks, posing significant challenges for many organizations. Cyber Threat Intelligence (CTI), which involves the collection, analysis, and dissemination of information about potential or current threats to an organization's cyber systems [36], can provide actionable insights to help organizations defend against these attacks. Large Language Models (LLMs) have the potential to revolutionize the field of CTI by enhancing the ability to process and analyze vast amounts of unstructured threat and attack data; allowing security analysts to utilize more intelligence sources than ever before. However, LLMs are prone to hallucinations [53] and misunderstandings of text, especially in specific technical domains, that can lead to a lack of truthfulness from the model [35]. This necessitates the careful consideration of using LLMs in CTI as their limitations can lead to them producing false or unreliable intelligence which could be disastrous if used to address real cyber threats.

The lack of proper benchmark tasks and datasets to evaluate LLM capabilities in CTI leaves their reliability and usefulness an open research question. Without standardized benchmarks, it is difficult to objectively measure and compare how effectively LLMs handle CTI tasks and generally understand the domain. General benchmarks like GLUE [52], SuperGLUE [51], MMLU [27], HELM [32], KOLA [54], and various domain-specific benchmarks [26, 55, 56] provide datasets and frameworks for evaluating LLMs in terms of general language understanding or domain-specific capabilities. However, these benchmarks fail to capture the practical aspects of cybersecurity. Limited works on LLM evaluations in cybersecurity [15, 16, 31, 33, 34] have primarily catered to specific cybersecurity

38th Conference on Neural Information Processing Systems (NeurIPS 2024) Track on Datasets and Benchmarks.

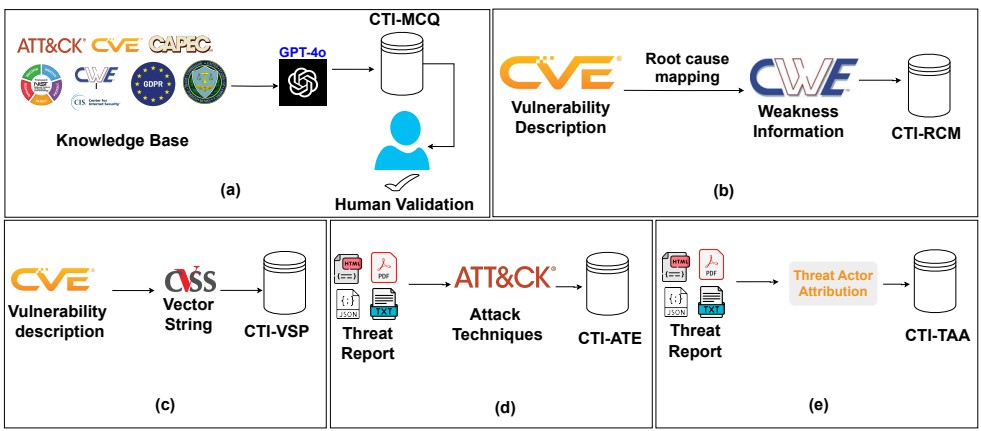

Figure 1: Overview of CTIBench.

industry or, focused on designing tasks that evaluate the memorization ability of LLMs, failing to capture the comprehension and problem-solving capabilities in the broad domain of CTI.

Addressing this gap, we propose **CTIBench**, a novel suite of benchmark tasks and datasets to evaluate LLMs in cyber threat intelligence. To this end, we construct a knowledge evaluation dataset, *CTI-MCQ*, comprising multiple-choice questions aimed at testing LLMs' understanding of crucial CTI concepts, including standards, threat identification, detection strategies, mitigation techniques, and best practices. We utilize various authoritative sources and standards within CTI domain such as NIST [29], MITRE [9], GDPR [1] to craft this dataset. In addition, we introduce three practical CTI tasks designed to evaluate LLMs' reasoning and problem-solving capabilities: (1) *CTI-RCM*, which involves mapping Common Vulnerabilities and Exposures (CVE) descriptions [40] to Common Weakness Enumeration (CWE) categories [41]; (2) *CTI-VSP*, which requires calculating Common Vulnerability Scoring System (CVSS) scores [23]; and (3) *CTI-ATE*, which focuses on extracting MITRE ATT&CK techniques from threat descriptions [14]. These tasks assess the LLMs' proficiency in understanding and evaluating cyber threats, vulnerabilities, and attack patterns. Furthermore, we propose a more complex task, *CTI-TAA*, where LLMs are tasked with analyzing publicly available threat reports and attributing them to specific threat actors or malware families. This task necessitates a thorough understanding of how different cyber threats or malware families have behaved in the past to identify meaningful correlations. Together, these tasks provide a robust framework for assessing LLMs in CTI. Figure 1 provides an overview of our benchmark. We evaluate five state-of-the-art LLMs—three commercial and two open-source—on these tasks. Our results and analysis provide important insights into the LLMs' capabilities in CTI analysis and highlight future avenues for research. We make the datasets and code publicly available.

Through CTIBench, we provide the research community with a robust tool to accelerate incident response by automating the triage and analysis of security alerts, enabling them to focus on critical threats and reducing response time. To the best of our knowledge, CTIBench is the first benchmark specifically designed to evaluate LLMs' comprehension, reasoning, and problem-solving abilities in the broad domain of CTI, addressing the limitations of existing benchmarks that either focus on general language understanding or specific cybersecurity tasks.

## 2 Related Work

**Large Language Models in Cybersecurity.** The advent of large language models (LLMs) like ChatGPT-3 [17], ChatGPT-4 [12], LLama models [49], Gemini [48] has enabled a wide range of applications across different domains, including cybersecurity. For example: code-based LLMs like CodeLlama [43] that can generate secure code are already integral parts of various industries. LLMs have also found applications in several other cybersecurity tasks like vulnerability detection [44, 22, 39], incidence response [13], program repair [45], IT operations [25], and cybersecurity knowledge assistance [47]. Despite these advancements of LLMs, there is a significant gap in their evaluation in the cybersecurity domain.

**Evaluating Large Language Models.** General evaluations benchmarks like GLUE [52], SuperGLUE [51], MMLU [27], HELM [32] and KOLA [54] assess general understanding capabilities of LLMs. Existing evaluation works in cybersecurity are either limited by their lack of comprehensiveness or being too narrow in their domain adaptation. For example: SECURE [16] proposes benchmarks for ICS industries, Purple Llama CyberSecEval [15], and SecLLMHolmes [50] evaluates LLMs' propensity to generate insecure code. SevenLLM [28] design tasks focused on extracting entities, relationships, and generating summaries categorized into understanding and generation tasks, and lack practical problem-solving evaluation of LLMs in the CTI domain. Other cybersecurity benchmarks [34] [31] [33] only evaluate the memorization and information extraction capabilities of LLMs.

# 3 CTIBench: A Benchmark for Evaluating LLMs in CTI

We are motivated by the need to create knowledge-intensive tasks to evaluate the cognitive capabilities of LLMs in Cyber Threat Intelligence (CTI) [30]. Our benchmark aims to verify that LLMs can understand, investigate, and analyze cyber threat reports. To this end, we have designed tasks and datasets that emphasize four fundamental cognitive capabilities of LLMs in CTI: *memorization* (ability to recall and utilize previously learned information), *understanding* (ability to comprehend the content and context), *problem-solving* (ability to apply knowledge and reasoning to address specific challenges), and *reasoning* (ability to draw logical conclusions and make informed decisions based on available information) [54].

## 3.1 CTI-MCQ: Cyber Threat Intelligence Multiple Choice Questions

**Data Collection.** To generate the CTI-MCQ dataset, we utilize various authoritative sources within the CTI domain. These sources include, among others, CTI frameworks such as NIST [29], the Diamond Model of Intrusion Detection [18], and regulations like GDPR [1]. Additionally, we incorporate CTI sharing standards such as STIX and TAXII [11] to formulate questions on fundamental cybersecurity and CTI knowledge. We leveraged the MITRE ATT&CK framework [46], CWE database [20], and CAPEC [19] to develop questions on attack patterns, threat actors, APT campaigns, detection methods, mitigation strategies, common software vulnerabilities and attack pattern enumeration. Finally, we supplement our dataset by manually collecting and curating questions from publicly available CTI quizzes, ensuring their relevance and accuracy by referencing established CTI resources. While these quizzes may introduce some bias, as they may not fully represent the entire spectrum of CTI knowledge, we ensured our diverse range of sources and manual curation process mitigated this potential limitation.

**Generating Questions Using GPT-4.** We utilize GPT-4o [5] to prepare the MCQs. Initially, we preprocess the content from the webpage to remove sections inappropriate for MCQ generation (such as page creation or update dates, references, etc.). We then optimize the prompt for creating questions that are challenging enough to test the knowledge of LLMs in cybersecurity. An example prompt is shown in Prompt A. We vary the number of questions we generate based on document length; we create more questions as the length of the document increases. We then randomly sample approximately 3000 questions for manual validation. This approach ensures a variety of questions while retaining quality

**Dataset Validation.** We manually analyze the quality of MCQs extracted from ChatGPT-4o to ensure the quality of the CTI-MCQ dataset. The human annotators were given access to the source URLs when analyzing the questions. We identified *two major issues from LLM-generated questions*: some questions had multiple correct options in the answers, and sometimes the LLM-given answer was incorrect. We removed the questions that were unanswerable from the given context or questions that included multiple correct options. We fixed the responses that had incorrect annotations provided by ChatGPT-4o.

Our final dataset consists of 2500 questions, out of which, 1578 questions are collected from MITRE, 750 from CWE, 40 from the manual collection, and 32 from standards and frameworks. This approach ensures a variety of questions while retaining quality.

## 3.2 CTI-RCM: Cyber Threat Intelligence Root Cause Mapping

Root cause mapping (RCM) identifies the underlying cause(s) of a vulnerability by correlating CVE records and bug tickets with CWE entries [20]. Accurate RCM is crucial for guides investments, policies, and practices to address and eliminate the root causes of vulnerabilities, benefiting both industry and government decision-makers. However, the current vulnerability management ecosystem does not perform this task accurately at scale [38], due to the complexity and nuance of CVE descriptions, the vast number of CWE categories (over 900 as of May 2024), and the need for domain expertise to interpret the relationships between them. To address this challenge, we designed the CTI-RCM task. Here is an example of RCM from MITRE [38]. The description for CVE-2018-15506 is as follows:

> In BubbleUPnP 0.9 update 30, the XML parsing engine for SSDP/UPnP functionality is vulnerable to an XML External Entity Processing (XXE) attack. Remote, unauthenticated attackers can use this vulnerability to: (1) Access arbitrary files from the filesystem with the same permission as the user account running BubbleUPnP, (2) Initiate SMB connections to capture a NetNTLM challenge/response and crack the cleartext password, or (3) Initiate SMB connections to relay a NetNTLM challenge/response and achieve Remote Command Execution in Windows domains.

The CWE is mapped to CWE-611: Improper Restriction of XML External Entity Reference. Here is the reasoning excerpt as shown in the reference link:

> Reasoning: Description says "vulnerable to an XML External Entity Processing (XXE) attack." There is additional information that focuses on technical impact, for instance, "attackers can do [X]", which is rarely useful for weaknesses, so that can be ignored. When you perform a text search on CWE for "XML External Entity Processing (XXE) attack" and "XXE," it returns CWE-611. When you click the entry, you see that the entry lists XXE as an "Alternate Term." Therefore, CWE-611 is the right mapping for this CVE.

As can be seen, this is a very nuanced task and requires a deep understanding of both the CVE descriptions and the CWE taxonomy to make accurate mappings.

**Data Collection.** For this task, we collect data from the National Vulnerability Database (NVD) [10]. The NVD database provides descriptions of past vulnerabilities identified by CVE, along with their associated mappings to Common Weakness Enumeration (CWE) entries. For our study, we specifically focus on vulnerabilities reported in the year 2024 that include associated CWE mappings. We then randomly sample 1,000 vulnerabilities to include in our dataset.

## 3.3 CTI-VSP: Cyber Threat Intelligence Vulnerability Severity Prediction

The Vulnerability Severity Prediction task involves predicting the Common Vulnerability Scoring System (CVSS) vector string from a given vulnerability description [23]. The CVSS is a standardized framework used to assess the severity of security vulnerabilities. It is composed of three metric groups: Base, Temporal, and Environmental. The Base metric group, which we focus on in our study, reflects the severity of a vulnerability based on its intrinsic characteristics. These characteristics are constant over time and assume a reasonable worst-case impact across different deployed environments. More details can be found in Appendix B.

While accurately calculating an accurate CVSS score requires additional detailed information such as original bug identification, third-party exploit analysis, or technical documentation for the vulnerable software, an approximation can be made using the initial CVE description[1]. The CVSS v3 Base Score is derived from the following eight metrics: *Attack Vector (AV)*, *Attack Complexity (AC)*, *Privileges Required (PR)*, *User Interaction (UI)*, *Scope (S)*, *Confidentiality Impact (C)*, *Integrity Impact (I)*, and *Availability Impact (A)*. Accurately calculating the CVSS score necessitates correctly mapping the vulnerability description to the appropriate severity levels for each metric. This task is inherently challenging due to the need for precise interpretation and understanding of technical language and context. Consequently, it serves as a robust benchmark for evaluating the capability of Large Language Models (LLMs) in understanding and processing cybersecurity-related information.

---

[1]https://www.first.org/cvss/v3.0/examples

Note that while there is a newer CVSS standard, CVSS 4.0, it was standardized in November 2023, and not all models might have adequate knowledge of the standard. Therefore, we focus on CVSS v3.

**Data Collection.** We use the same data source as the CTI-RCM for this task. Specifically, we collect 1,000 vulnerability descriptions from 2024 and their associated CVSS v3 strings.

## 3.4 CTI-ATE: Cyber Threat Intelligence Attack Technique Extraction

The Attack Technique Extraction task involves identifying specific attack patterns from a given description of threat behavior and mapping them to the corresponding MITRE ATT&CK technique IDs [14]. These technique IDs represent distinct adversarial methods used at various stages of an attack lifecycle.

Consider the following example[2]:

> Janicab is an OS X trojan that exploited a valid Apple Developer ID to deceive users during installation. It captured both audio and screenshots, which were then transmitted to a remote command and control (C2) server. For persistence, Janicab employed a cron job on affected Mac devices. The use of a legitimate Apple Developer ID enabled the trojan to bypass security restrictions by signing the malicious code.

From this description, we can identify the following relevant attack technique IDs: (i) T1123 – Audio Capture, (ii) T1053 – Scheduled Task, (iii) T1113 – Screen Capture, and (iv) T1553 – Subvert Trust Controls.

This task is valuable for CTI practitioners, as accurately mapping observed behaviors to the corresponding MITRE ATT&CK techniques is essential for designing effective mitigation strategies and deploying targeted security measures. By linking specific behaviors to their respective technique IDs, security teams can better understand the adversary's tactics, techniques, and procedures (TTPs), enabling them to take proactive actions to disrupt or mitigate ongoing threats.

**Data Collection.** For this task, we curated a dataset using information from the MITRE ATT&CK framework [14], which provides comprehensive descriptions of various malware and their associated adversarial behaviors, each categorized by a unique attack technique ID based on open-source threat reports. Our dataset includes 30 instances of malware reported in 2024, alongside their corresponding attack technique IDs—information that extends beyond the knowledge cutoff of the LLMs under evaluation. We also supplemented the dataset with 30 malware instances reported before 2024. For this task, we focused solely on techniques (excluding sub-techniques). In total, the dataset comprises 397 unique attack techniques.

## 3.5 CTI-TAA: Cyber Threat Intelligence Threat Actor Attribution

Threat actor attribution is a crucial process of identifying the individuals, groups, or organizations responsible for a cyberattack or malicious activity. This is usually done by analyzing various indicators of compromise (IOCs), such as malware signatures, attack vectors, infrastructure, tactics, techniques, procedures (TTPs), and other contextual information like geopolitical motives or previous attack patterns. This is a challenging task because threat actors often use sophisticated evasion tactics, shared tools and techniques, limited and incomplete data, rapidly changing TTPs, and inherent biases in analysis [37]. This task exemplifies abductive reasoning, which involves forming plausible conclusions from incomplete information, often seeking the best explanation [21]. By evaluating LLMs on this task, we aim to benchmark their capability to perform complex reasoning and analysis in the context of CTI.

**Data Collection** We create a small-scale dataset to enable LLMs to reason about this intricate CTI concept. To this end, we collect 50 threat reports from reputed vendors that have an Advanced Persistent Threat (APT) group attributed to the reports [24]. The reports vary in the amount of detail provided about the threat actor.

To create a controlled evaluation setting, we remove all mentions of the threat actors and their associated malware campaign names, replacing them with placeholders. We then task the LLMs with

---

[2]https://attack.mitre.org/software/S0163/

attributing the reports to known threat actors. To further ensure accuracy, we manually verify each LLM response to account for the multiple aliases that threat actors often use.

# 4 Experiments and Results

## 4.1 Experimental Settings

We evaluate ChatGPT-3.5 (`gpt-3.5-turbo`) [3], ChatGPT-4 (`gpt-4-turbo`) [4], Gemini-1.5 [2], LLAMA3-70B [7] and LLAMA3-8B [8] on our benchmark tasks. We set the temperature of LLMs at 0 and $top\_p = 1$ to obtain more deterministic responses. Each task is evaluated on a zero-shot prompt template with instruction of LLMs to act as a cybersecurity expert. Below, we show a prompt template used for the vulnerability root cause mapping. Please see Appendix C for evaluation prompts used for all tasks.

> You are a cybersecurity expert specializing in cyber threat intelligence. Analyze the following CVE description and map it to the appropriate CWE. Provide a brief justification for your choice. Ensure the last line of your response contains only the CWE ID.
>
> CVE Description:
> $\{description\}$

## 4.2 Evaluation Metrics

We use accuracy to evaluate the CTI-MCQ and CTT-RCM tasks, as both tasks are equivalent to multi-class classification. For the CTI-VSP task, we compute the mean absolute deviation (MAD) between the CVSS v3.1 scores of the ground truth and the model's predictions. Although we ask the model to predict a vector string, the CVSS score can be deterministically derived from it. We utilize the Python library *cvss* [42] to compute the CVSS score (a numerical value in the range of 0-10 that determines the overall severity of a vulnerability) from the predicted string, ensuring that any potential errors from the LLM performing the computation are eliminated. This approach focuses solely on assessing the LLM's reasoning ability regarding vulnerability. We adopt the Micro-F1 score as the evaluation metric for the CTI-ATE task. Given that this task requires accurately extracting the relevant attack techniques from the provided text and mapping them to their corresponding MITRE ATT&CK technique IDs, the Micro-F1 score is suitable for capturing both precision and recall in a multi-label classification setting.

For the CTI-TAA task, we categorize the predictions into three types: *correct answer* (when the LLM accurately identifies the threat actor or one of its aliases), *plausible answer* (when the threat report lacks sufficient details to pinpoint the answer, but the LLM provides a plausible or related threat actor within a similar group), and *incorrect answer* (when the LLM misattributes the threat actor due to misjudgment, hallucination, or spurious correlation). Based on these categories, we compute two types of accuracy: Correct Accuracy, which is the fraction of correct answers, and Plausible Accuracy, which is the fraction of correct and plausible answers combined. Details on generating the ground truth and evaluation for CTI-TAA are provided in Appendix D.

## 4.3 Results Summary

Table 1 presents the performance of various LLMs on our benchmark CTI tasks. The results indicate that GPT-4 outperforms other models across all tasks except for CTI-VSP, where Gemini-1.5 takes the lead. Despite being open-source, LLAMA3-70B performs comparably to Gemini-1.5 and even outperforms it on three tasks, though it struggles with the CTI-VSP task. ChatGPT-3.5 exceeds the performance of LLAMA3-8B but is generally outperformed by the other models across most tasks. LAMA3-8B, being a smaller model, fails to match the performance of larger models on tasks requiring more nuanced understanding and reasoning. However, it performs decently on the CTI-MCQ task.

| Model | CTI-MCQ (Acc) | CTI-RCM (Acc) | CTI-VSP (MAD) | CTI-ATE (Macro-F1) | CTI-TAA (Acc) | |
| --- | --- | --- | --- | --- | --- | --- |
| | | | | | Correct | Plausible |
| ChatGPT-4 | **71.0** | **72.0** | 1.31 | **0.6388** | **52** | **86** |
| ChatGPT-3.5 | 54.1 | 67.2 | 1.57 | 0.3108 | 44 | 62 |
| Gemini-1.5 | 65.4 | 66.6 | **1.09** | 0.4612 | 38 | 74 |
| LLAMA3-70B | 65.7 | 65.9 | 1.83 | 0.4720 | 52 | 80 |
| LLAMA3-8B | 61.3 | 44.7 | 1.91 | 0.1562 | 28 | 36 |

Table 1: Results of different LLMs on the benchmark datasets (bold indicates the best performing model, lower is better for MAD)

## 5 Analysis

### 5.1 CTI-MCQ

The heatmap analysis of error correlations, in Figure 2(a), shows that the larger models exhibit higher error correlations. This trend suggests that these models, such as ChatGPT-4, Gemini-1.5, and LLAMA3-70B, are likely to make similar mistakes when answering the CTI-MCQ. For instance, ChatGPT-4 shows error correlations of 0.52 with Gemini-1.5 and 0.55 with LLAMA3-70B, while Gemini-1.5 and LLAMA3-70B have a correlation of 0.54. In Figure 2(b), we show the number of questions incorrectly answered by a number of LLMs. Overall, 293 questions were answered incorrectly by all the models (5) in the evaluation. Upon inspection, we found these questions to be related to mitigation plans and tools and techniques used by adversaries. We show a sample of such questions in Appendix Table 5.

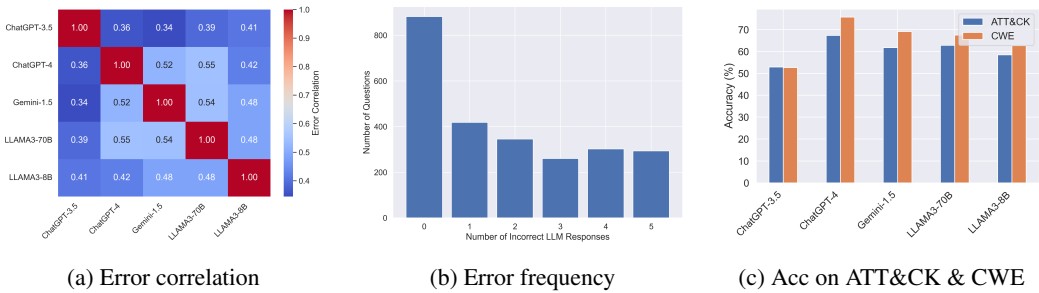

| (a) Error correlation | (b) Error frequency | (c) Acc on ATT&CK & CWE |
| --- | --- | --- |

Figure 2: Error analysis on the CTI-MCQ tasks

Figure 2(c) displays the accuracy of different LLMs on multiple-choice questions (MCQs) from two primary sources: MITRE ATT&CK and CWE. Given that MITRE ATT&CK information is more volatile compared to the more stable nature of CWEs, models generally perform better on questions sourced from CWE. However, even the best-performing model, ChatGPT-4, achieves an accuracy of 75.65%, indicating that there is still further room for improvement.

While Table 1 presents the results for the MCQ task without an explicit reasoning step, we also performed an additional evaluation incorporating an explicit reasoning prompt. The detailed results of this evaluation are provided in Appendix E. However, this approach did not consistently lead to performance improvements. We hypothesize that this is because the MCQ task primarily relies on memorization rather than reasoning.

### 5.2 CTI-RCM

In the CTI-RCM task, LLMs are assigned to identify the underlying cause(s) of a vulnerability by correlating CVE Records with CWE entries. Figure 3(a) shows the frequency distribution of word counts across the CVE descriptions in our dataset, revealing a right-skewed distribution where most descriptions have a lower word count. Figure 3(b) demonstrates that all models, except LLAMA3-8B, improve accuracy with longer descriptions, peaking in the 54-69 word range. This trend suggests that longer descriptions offer more context, aiding the models in accurately mapping CVE records to CWE entries. However, performance declines for the longest CVE descriptions. The most likely reason is

that as description length increases, the potential for matching multiple weaknesses increases, causing LLMs to struggle to identify the most appropriate one.

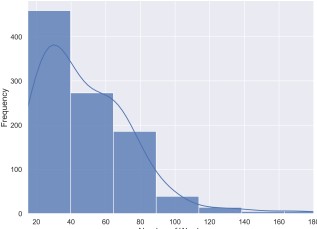

(a) Distribution of word counts in CVE descriptions

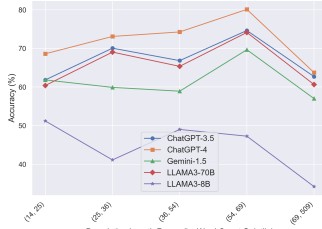

(b) Accuracy vs. description length for different LLMs for CTI-RCM

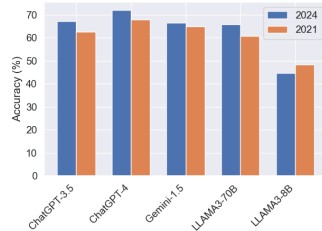

(c) Accuracy for the CTI-RCM task for 2 different years

Figure 3: CTI-RCM and CTI-VSP analysis

When creating the CTI-RCM dataset, we gathered CVE descriptions exclusively from 2024, which is beyond the training cut-off date for the models we evaluated. To investigate model performance further, we conducted an additional experiment using CVE descriptions and their associated CWE mappings from 2021. The results, presented in Figure 3 (c), show that four out of five models perform slightly worse on the 2021 dataset. This suggests the task is inherently challenging and could serve as a robust evaluation benchmark for future LLMs.

## 5.3 CTI-VSP

In the CTI-VSP task, LLMs are tasked with predicting the CVSS v3 Base String based on the CVE description, which is then converted to a CVSS score using a predefined formula. We evaluate the performance of the LLMs by computing the Mean Absolute Deviation (MAD) from the ground truth in the dataset. Like the CTI-RCM task, models tend to perform better with longer descriptions, as indicated by a decrease in MAD in Figure 3(c). However, there is a noticeable performance drop for all the models, as evidenced by the sharp increase in MAD for the last quintiles. This pattern suggests that while longer descriptions provide more context and generally improve performance, they can also introduce complexity that leads to misattribution of the severity of the threat. This combined finding from the CTI-VSP and CTI-RCM tasks indicates that additional description length does not necessarily equate to better performance and may hinder the models' ability to assess the threat accurately.

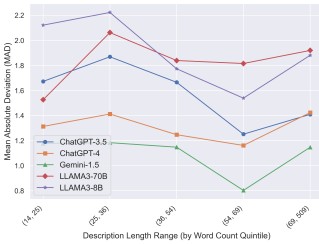

(a) MAD vs. description length for different LLMs for CTI-VSP

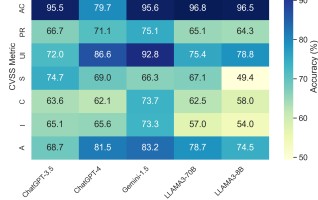

(b) Accuracy with respect to base metrics

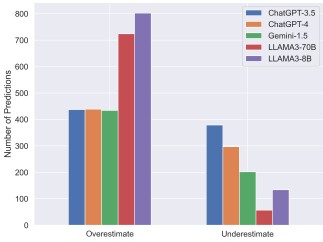

(c) Error analysis of severity score prediction

Figure 4: CTI-VSP analysis

We illustrate the accuracy of LLMs in predicting the CVSS v3 Base Metrics in Figure 4(a). The CVSS Base Metrics include Attack Vector (AV), Attack Complexity (AC), Privileges Required (PR), User Interaction (UI), Scope (S), Confidentiality Impact (C), Integrity Impact (I), and Availability Impact (A) (detailed in Appendix B). The heatmap shows that all models perform relatively well predicting AV, AC, and UI. However, they struggle with PR, S, C, and I, suggesting that CVE descriptions often lack sufficient detail to infer these metrics accurately.

Figure 4(b) illustrates the number of overestimations and underestimations made by LLMs when predicting the CVSS base score from a vulnerability description. Overestimation occurs when the predicted score is higher than the actual score, while underestimation occurs when the predicted score is lower. All models exhibit a higher frequency of overestimation compared to underestimation, with this tendency being particularly pronounced in the two open-source LLAMA models. This suggests that LLMs may need calibration to improve their accuracy in threat severity prediction.

## 5.4 CTI-ATE

The results in Table 1 demonstrate that ChatGPT-4 significantly outperforms other models on the CTI-ATE task, underscoring the complexity of the task. To further investigate model performance, we evaluated the models on samples collected before and after their respective knowledge cutoff dates. These results are presented in Table 2. Most models exhibit slightly better performance on samples collected before their knowledge cutoff, except for Gemini, which performs better on post-cutoff samples. The performance differences are generally insignificant, suggesting that this can serve as a reasonable indicator of how LLMs will fare on future threat reports.

| Model | Before (F1) | After (F1) |
|---|---|---|
| ChatGPT-4 | 0.6542 | 0.6208 |
| ChatGPT-3.5 | 0.3420 | 0.3333 |
| Gemini-1.5 | 0.4360 | 0.5263 |
| LLAMA3-70B | 0.4934 | 0.4297 |
| LLAMA3-8B | 0.1813 | 0.1366 |

Table 2: Performance comparison on the CTI-ATE task, evaluated on samples before and after the models' knowledge cutoff dates.

## 5.5 CTI-TAA

The results of the threat actor attribution tasks (Table 1) indicate that LLMs can perform nuanced analyses of the information presented in threat reports and make insightful correlations. While smaller models like LLAMA3-8B struggle with more complex reasoning tasks, larger models demonstrate reasonable performance. Our analysis of the reasoning provided by LLMs suggests that they possess a general understanding of the cyber threat landscape, though they may occasionally misattribute information. Below, we present representative examples of threat-attribution predictions made by ChatGPT-4.

**Correct response:** We task the LLM to predict the threat actor CHRYSENE given a threat report by replacing the mention of the threat actor and its campaign with [PlaceHolder].[3] CHRYSENE is an Iranian cyberespionage group active since 2014, targeting Middle Eastern governments and various industries. ChatGPT-4 predicted the threat actor as OilRig, which is an alias of CHRYSENE.

**Plausible response:** We task the LLM to predict the threat actor MuddyWater given a threat report by replacing the mention of the threat actor and its campaign with [PlaceHolder].[4] MuddyWater is a cyber espionage group linked to Iran's Ministry of Intelligence and Security (MOIS) that targets government and private sectors across the Middle East, Asia, Africa, Europe, and North America. ChatGPT-4 predicted the threat actor as APT35, which is not the alias of MuddyWater but shares some common attack patterns like originating from Iran, targeting the Middle East and North America, and using multi-stage attacks.

**Incorrect response:** We task the LLM to predict the threat actor APT41 given a threat report by replacing the mention of the threat actor and its campaign with [PlaceHolder].[5] APT41 is an espionage group from China that has been active since 2012 and involved in financially motivated operations, targeting healthcare, telecom, technology, and video game industries. ChatGPT-4 incorrectly predicted the threat actor as APT29, based on the fact that APT29 has been active since 2012 and its state-sponsored espionage operations.

---

[3]https://www.welivesecurity.com/en/eset-research/oilrigs-outer-space-juicy-mix-same-ol-rig-new-drill-pipes/

[4]https://www.deepinstinct.com/blog/darkbeatc2-the-latest-muddywater-attack-framework

[5]https://www.trendmicro.com/en_us/research/24/d/earth-freybug.html

**Impact of Knowledge Cutoff:** We evaluate the LLMs on the CTI-TAA task using datasets from both before and after their knowledge cutoff dates. The results are displayed in the table below. With the exception of LLAMA3-8B, all other LLMs demonstrated better performance on datasets available during their training period, suggesting that memorization plays a role in improving performance to some extent in this task.

| Model | Before (Acc) | | After (Acc) | |
|---|---|---|---|---|
| | **Correct** | **Plausible** | **Correct** | **Plausible** |
| ChatGPT-4 | 58.06 | 90.32 | 42.10 | 78.95 |
| ChatGPT-3.5 | 50.00 | 75.00 | 43.48 | 60.87 |
| Gemini-1.5 | 34.61 | 76.92 | 41.66 | 70.83 |
| LLAMA3-70B | 58.06 | 83.87 | 42.10 | 73.68 |
| LLAMA3-8B | 25.00 | 25.00 | 28.94 | 39.47 |

Table 3: Performance comparison of models before and after knowledge cutoff dates for the CTI-TAA task.

## 5.6 Compute Cost

Appendix G presents the detailed token counts for each task. GPT-4 generated significantly longer responses across all tasks than the other models.

## 6 Ethical Concerns

All the evaluation tasks in our proposed benchmark utilize publicly available threat information from reputable sources such as NIST, MITRE, CVE, CWE, and EU. None of the datasets include personal information or make sensitive judgments related to social issues, bias, deception, or discrimination.

## 7 Limitations

While our evaluation of large language models (LLMs) for Cyber Threat Intelligence (CTI) tasks provides valuable insights, it is important to recognize certain limitations. First, the extensive scope of CTI activities presents a significant challenge, and our study has focused on a limited subset of tasks to assess the capabilities of LLMs. This selection may not fully capture the breadth of functionality required for comprehensive CTI operations. In future work, we plan to expand the range of evaluated tasks to encompass a broader spectrum of CTI activities, thereby ensuring a more holistic assessment of LLM performance in this domain.

Second, our evaluation is restricted to English-language CTI techniques. This limitation neglects the global nature of cyber threats, which frequently span multiple languages and regions. To address this, future studies will incorporate multilingual CTI evaluations, better reflecting the diverse linguistic landscape of cyber threats and improving the applicability of LLMs in international cybersecurity contexts.

Additionally, there is a genuine risk of the malicious use of LLMs to exploit CTI knowledge for harmful purposes. For instance, if misused, LLMs could generate and disseminate compelling but false threat intelligence reports. Such reports might mislead decision-makers, result in the misallocation of resources, or prompt inappropriate responses. Benchmarking the potential for such misuse remains an open area for future research.

## 8 Conclusion

The emergence of LLMs has opened up new possibilities in cybersecurity, especially in Cyber Threat Intelligence (CTI). However, their capabilities and limitations in this domain remain unclear. In this paper, we introduce CTIBench, a benchmark designed to evaluate LLM performance in various CTI tasks. Our evaluation offers valuable insights into the knowledge and capabilities of LLMs across various CTI aspects, as well as their limitations. We aim for our benchmark to help researchers understand the practical applications of LLMs in CTI and to pave the way for their reliable use and the effective detection and mitigation of cyberthreats.

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

# Appendix

## A  Prompt for generating CTI-MCQ

**Prompt for generating MCQs with ChatGPT-4o from MITRE ATT&CK**
(Italized parts are unique to MITRE ATT&CK)

You are a cybersecurity expert specializing in cyber threat intelligence. Given the text below, please generate a maximum of 5 multiple-choice questions with four possible options each.

Follow these requirements:
1. Question Format: Each question must have four options. The options should be challenging and require careful consideration. Avoid creating options that could be interpreted as correct under different circumstances.
2. Target Audience: The questions should be suitable for security professionals with three to five years of experience in cyber threat intelligence. *Avoid generic questions such as "What is the objective?", "Which operating system can be targeted?".*
3. Content Coverage: Aim to cover various sections of the document to ensure a comprehensive evaluation of the candidate's knowledge. Include context-specific questions that require an understanding of the document's content.
4. Technical Precision: Use precise terminology and concepts relevant to cyber threat intelligence. Incorporate situational or scenario-based questions where applicable.
5. *Include Technique IDs and Names: Ensure that all questions, where applicable, mention both the ID and the full name of the MITRE ATT&CK pattern technique.*
6. *Premise Inclusion: Each question should include a premise indicating it pertains to MITRE ATT&CK, specifying the relevant platform (Enterprise, ICS, or Mobile) where necessary.*
7. Output Format: Return the output in TSV format (must be tab-separated) with the following columns: Question, Option A, Option B, Option C, Option D, Correct Answer (A, B, C, D), and Explanation.

**Important:** Only return the TSV content as specified. Do not include any additional text or commentary outside the TSV format.

Text:
...

## B  CVSS Base Metric

The Base Metrics group in CVSS encapsulates a vulnerability's fundamental and immutable properties and is crucial for understanding the potential impact and exploitability of a vulnerability [23]. It consists of the following metrics:

1. **Attack Vector (AV):** The Attack Vector metric evaluates the proximity an attacker must have to the vulnerable component to exploit it. Possible values:

   - *Network (N):* The attacker can exploit the vulnerability remotely over a network. This typically indicates the highest risk, as remote exploitation can be conducted without physical access.
   - *Adjacent (A):* The attacker requires access to the local network or adjacent hardware. Examples include attacks over Bluetooth or local subnet.
   - *Local (L):* The attacker needs local access to the system, either physically or via a local account. This increases the difficulty compared to network attacks.
   - *Physical (P):* The attacker must have physical contact or access to the target system, which makes this the most challenging attack vector.

2. **Attack Complexity (AC):** This metric measures the conditions beyond the attacker's control that must exist to exploit the vulnerability. Possible values:

   - *Low (L):* Exploiting the vulnerability does not require any special conditions.
   - *High (H):* Exploitation requires specific conditions or configurations.

3. **Privileges Required (PR):** Privileges Required assesses the level of access or privileges an attacker must have to successfully exploit the vulnerability. Possible values:

- *None (N):* The attacker does not need any privileges; they can exploit the vulnerability as an unauthenticated user. This typically results in a higher score since the barrier to exploitation is minimal.
- *Low (L):* The attacker needs basic user privileges. This implies that some level of access is required, but not necessarily elevated privileges.
- *High (H):* Exploitation requires administrative or high-level privileges. This makes exploitation significantly harder.

4. **User Interaction (UI):** User Interaction measures the degree to which a user must participate in the attack. Possible values:

- *None (N):* The attacker can exploit the vulnerability without any interaction from the user. This indicates a higher risk as the attack can be automated and spread more easily.
- *Required (R):* Successful exploitation requires user interaction, such as clicking a link or opening a file. This adds a layer of difficulty for the attacker since it relies on social engineering or user behavior.

5. **Scope (S):** Scope evaluates whether a vulnerability in one component impacts resources beyond its security scope. This metric assesses the potential broader implications of an exploit. Possible values:

- *Unchanged (U):* The vulnerability only affects resources within the same security scope. This typically means the impact is contained within a single component or system.
- *Changed (C):* Exploitation of the vulnerability impacts resources beyond the vulnerable component's security scope.

6. **Confidentiality Impact (C):** Confidentiality Impact measures the extent to which information disclosure can occur due to a successful exploit. Possible values:

- *None (N):* There is no impact on confidentiality; no sensitive information is disclosed.
- *Low (L):* Access to some information is gained, but the attacker cannot control what is disclosed, or the information is not highly sensitive.
- *High (H):* Complete disclosure of all data on the affected system. This indicates a severe impact, potentially exposing highly sensitive information.

7. **Integrity Impact (I):** Integrity Impact assesses the extent to which data can be altered or tampered with by an attacker. Possible values:

- *None (N):* There is no impact on the integrity of the data or system.
- *Low (L):* Data can be modified, but the extent is limited or the attacker does not have control over the modifications.
- *High (H):* The attacker can make extensive or significant modifications to the data, potentially compromising the integrity of the entire system or dataset.

8. **Availability Impact (A):** Availability Impact measures the potential disruption to the availability of the affected component due to a successful exploit.

- *None (N):* There is no impact on availability; the system remains fully operational.
- *Low (L):* Performance is degraded, or there are occasional disruptions, but the system remains available.
- *High (H):* The system is completely unavailable or severely degraded, indicating a significant impact on availability and potentially causing a denial of service.

# C   Evaluation Prompts

## C.1   CTI-MCQ: Cyber Threat Intelligence Multiple Choice Questions

**Prompt used for LLM Evaluation**   We use the following prompt to generate answers from LLMs for the MCQs

You are a cybersecurity expert specializing in cyber threat intelligence. You are given a multiple-choice question (MCQ) from a Cyber Threat Intelligence (CTI) knowledge benchmark dataset. Your task is to choose the best option among the four provided. Return your answer as a single uppercase letter: A, B, C, or D.

**Question:**
{*question*}

**Options:**
A) {*option_a*}
B) {*option_b*}
C) {*option_c*}
D) {*option_d*}

**Important:** The last line of your answer should contain only the single letter corresponding to the best option, with no additional text.

## C.2 CTI-RCM: Cyber Threat Intelligence Root Cause Mapping

**Prompt used for LLM Evaluation**    We use the following prompt to identify the vulnerability root cause mapping:

You are a cybersecurity expert specializing in cyber threat intelligence. Analyze the following CVE description and map it to the appropriate CWE. Provide a brief justification for your choice. Ensure the last line of your response contains only the CWE ID.

CVE Description:

## C.3 CTI-VSP: Cyber Threat Intelligence Vulnerability Severity Prediction

**Prompt used for LLM Evaluation**    We use the following prompt to predict CVSS v3.1 string using LLM:

Analyze the following CVE description and calculate the CVSS v3.1 Base Score. Determine the values for each base metric: AV, AC, PR, UI, S, C, I, and A. Summarize each metric's value and provide the final CVSS v3.1 vector string.

Valid options for each metric are as follows:
- Attack Vector (AV): Network (N), Adjacent (A), Local (L), Physical (P)
- Attack Complexity (AC): Low (L), High (H)
- Privileges Required (PR): None (N), Low (L), High (H)
- User Interaction (UI): None (N), Required (R)
- Scope (S): Unchanged (U), Changed (C)
- Confidentiality (C): None (N), Low (L), High (H)
- Integrity (I): None (N), Low (L), High (H)
- Availability (A): None (N), Low (L), High (H)

Summarize each metric's value and provide the final CVSS v3.1 vector string. Ensure the final line of your response contains only the CVSS v3 Vector String in the following format:

Example format: CVSS:3.1/AV:N/AC:L/PR:N/UI:N/S:U/C:H/I:H/A:H

CVE Description:

## C.4 CTI-ATE: Cyber Threat Intelligence Attack Technique Extraction

**Prompt used for LLM Evaluation**    We provide the list of all Enterprise attack techniques in the prompt for the CTI-ATE task:

> You are a cybersecurity expert specializing in cyber threat intelligence. Extract all MITRE Enterprise attack patterns from the following text and map them to their corresponding MITRE technique IDs. Provide reasoning for each identification. Ensure the final line contains only the IDs for the main techniques, separated by commas, excluding any subtechnique IDs. MITRE Enterprise IDs are given below as reference.
>
> **Text:**
>
> **List of All MITRE Enterprise technique IDs**

### C.5 CTI-TAA: Cyber Threat Intelligence Threat Actor Attribution

**Prompt used for LLM Evaluation**  We use the following prompt for the threat actor attribution task:

> You are a cybersecurity expert specializing in cyber threat intelligence. You are given a threat report that describes a cyber incident. Any direct mentions of the threat actor group, specific campaign names, or malware names responsible have been replaced with [PLACEHOLDER]. Your task is to analyze the report and attribute the incident to a known threat actor based on the techniques, tactics, procedures (TTPs), and any other relevant information described. Please provide the name of the threat actor you believe is responsible and briefly explain your reasoning.
> Threat Report:

## D CTI-TAA Evaluation

To evaluate the threat actor attribution task, we crafted the CTI-TAA dataset. The ground truth for the *correct answer* was established by extracting information directly from original documents, ensuring an exact match with named entities. Additionally, we collected aliases associated with the threat actors using Malpedia [6]. To ensure comprehensive coverage of possible aliases, we referenced the individual Malpedia pages of the threat actors and explored alias pages, capturing secondary aliases derived from primary ones that were not initially included.

We consider all identified aliases as equivalent to the ground truth. Furthermore, we included all related threat actors for the original threat actors, sourced from related or associated groups' information on the MITRE website. This forms the initial plausible set of actors. Given that an alias of one threat actor may sometimes be an alias of another, creating a chain due to incomplete alias lists on individual pages, and considering that related actors may have their aliases, we employed a graph-searching algorithm to determine the accuracy of model predictions.

We used a breadth-first search (BFS) algorithm to verify correctness, starting from the LLM response and traversing only nodes connected by an alias. The response is marked as correct if a match with the ground truth is found before the search is exhausted. We considered nodes connected via both aliases and related group nodes to identify related groups. The response is considered related if a match with the ground truth is found through this expanded search. If no matches are found, the response is deemed incorrect.

The algorithm is shown in Algorithm 1. We include the algorithm in our released code.

## E CTI-MCQ with reasoning

We re-evaluated the LLMs on the MCQ task by modifying the prompt to explicitly instruct the models to reason through the problem before providing an answer. Table 4 presents the results. While explicit reasoning prompts substantially increased computational costs, they generally do not translate into improved performance. As shown in the table, only GPT-3.5 exhibited a notable performance boost, while for LLAMA3-8B, the reasoning prompt led to a decline in performance.

**Algorithm 1** Evaluate Model's Prediction for the CIT-TAA task

---

**Require:** LLM response, Ground truth entities, Alias and related actor mappings
**Ensure:** Correctness or plausibility of the response
 1: Initialize **correct** = False, **related** = False
 2: **Check for Correctness**
 3: Initialize BFS queue with LLM response node
 4: **while** queue is not empty and **correct** is False **do**
 5:     Dequeue a node
 6:     **if** node matches ground truth **then**
 7:         **correct** = True
 8:     **else**
 9:         Enqueue all alias-connected nodes
10:     **end if**
11: **end while**
12: **Check for Plausibility**
13: **if correct** is False **then**
14:     Initialize BFS queue with LLM response node
15:     **while** queue is not empty and **related** is False **do**
16:         Dequeue a node
17:         **if** node matches ground truth **then**
18:             **related** = True
19:         **else**
20:             Enqueue all alias and related group-connected nodes
21:         **end if**
22:     **end while**
23: **end if**
24: **Determine Result**
25: **if correct then**
26:     Response is **Correct**
27: **else if related then**
28:     Response is **Related**
29: **else**
30:     Response is **Incorrect**
31: **end if**

---

| Model | Acc Without Reasoning | Acc With Reasoning |
|-------|----------------------|--------------------|
| ChatGPT-4 | 71.00 (105) | 71.84 (618) |
| ChatGPT-3.5 | 54.08 (10) | 59.16 (227) |
| Gemini-1.5 | 65.44 (5) | 65.68 (360) |
| LLAMA3-70B | 65.72 (1) | 65.48 (230) |
| LLAMA3-8B | 61.32 (5) | 55.80 (269) |

Table 4: Comparison of model accuracy with and without reasoning. Numbers in parentheses indicate the approximate number of tokens generated (in thousand tokens)

## F   CTI-MCQ Samples

We show a sample of incorrectly answered MCQs by LLMs from CTI-MCQ dataset in Table 5.

## G   Compute Cost

We show the approximate number of tokens used in Prompts and in LLM responses in Table 6.

Table 5: A sample of questions incorrectly answered by all the LLMs under evaluation

| Question Type | Question | Option A | Option B | Option C | Option D | Correct Answer |
|---|---|---|---|---|---|---|
| Mitigation | Under the MITRE ATT&CK framework for Enterprise, which mitigation can help prevent adversaries from creating or interacting with system services using a lower permission level? | Behavior Prevention on Endpoint | Malware Detection | Privileged Account Management | Restrict File & Directory Permissions | C |
| Tool | In the context of MITRE ATT&CK for Enterprise, which of the following tools can use PowerShell to discover email accounts as per T1087.003 Account Discovery: Email Account? | TrickBot | MailSniper | Magic Hound | Lizar | C |
| Method | What is a common method used by adversaries for outbound traffic in Technique ID T1102.002? | Using DNS tunneling | Sending emails to command servers | Making HTTP requests to compromised blogs | Using FTP to upload data | C |
| Malware | Which of the following malware examples triggers on a magic packet in TCP or UDP packets? | BUSHWALK | Ryuk | SYNful Knock | Penquin | D |
| Detection | How does Clop avoid sandbox detection? | Using GetTickCount function | Disabling system clock Scheduled Task/Job | Calling NtDelayExecution | Using the sleep command | D |

| Model | CTI-MCQ | CTI-RCM | CTI-VSP | CTI-ATE | CTI-TAA |
|---|---|---|---|---|---|
| Prompt | 419 | 143 | 454 | 102 | 149 |
| ChatGPT-4 | 105 | 269 | 485 | 30 | 32 |
| ChatGPT-3.5 | 10 | 135 | 170 | 18 | 21 |
| Gemini-1.5 | 5 | 100 | 344 | 27 | 32 |
| LLAMA3-70B | 1 | 140 | 442 | 22 | 25 |
| LLAMA3-8B | 5 | 136 | 365 | 36 | 21 |

Table 6: Approximate number of tokens for Prompt and LLM response for each task (in thousands of token units)

