# Supplementary material for CTIBench: A Benchmark for Evaluating LLMs in Cyber Threat Intelligence

## 1    Dataset Documentations

### 1.1    Hosted URLs

**Huggingface:** `https://huggingface.co/datasets/AI4Sec/cti-bench`

**Github:** `https://github.com/xashru/cti-bench/tree/main/data`

**Croissant:** `https://huggingface.co/api/datasets/AI4Sec/cti-bench/croissant`

### 1.2    Dataset description

CTIBench is a comprehensive suite of benchmark tasks and datasets designed to evaluate LLMs in the field of CTI. It consists of the following components:

1. *CTI-MCQ:* A knowledge evaluation dataset with multiple-choice questions to assess the LLMs' understanding of CTI standards, threats, detection strategies, mitigation plans, and best practices. This dataset is built using authoritative sources and standards within the CTI domain, including NIST, MITRE, and GDPR.

2. *CTI-RCM:* A practical task that involves mapping Common Vulnerabilities and Exposures (CVE) descriptions to Common Weakness Enumeration (CWE) categories. This task evaluates the LLMs' ability to understand and classify cyber threats.

3. *CTI-VSP:* Another practical task that requires calculating the Common Vulnerability Scoring System (CVSS) scores. This task assesses the LLMs' ability to evaluate the severity of cyber vulnerabilities.

4. *CTI-TAA:* A task that involves analyzing publicly available threat reports and attributing them to specific threat actors or malware families. This task tests the LLMs' capability to understand historical cyber threat behavior and identify meaningful correlations.

### 1.3    Dataset structure

The dataset consists of 5 TSV files, each corresponding to a different task. Each TSV file contains a "Prompt" column used to pose questions to the LLM. Most files also include a "GT" column that contains the ground truth for the questions, except for "cti-taa.tsv". The evaluation scripts for the different tasks are available in the associated GitHub repository.

### 1.4    Dataset creation

#### 1.4.1    Rationale

The lack of proper benchmark tasks and datasets to evaluate LLM capabilities in CTI leaves their reliability and usefulness an open research question. This dataset was curated to evaluate the ability

Submitted to the 38th Conference on Neural Information Processing Systems (NeurIPS 2024) Track on Datasets and Benchmarks. Do not distribute.

of LLMs to understand and analyze various aspects of open-source CTI. These datasets evaluate the reasoning, understanding and problem-solving abilities of LLMs in cyber-threat intelligence. To the best of our knowledge, CTIBench is the first benchmark specifically designed to evaluate LLMs' comprehension, reasoning, and problem-solving abilities in the broad domain of CTI, addressing the limitations of existing benchmarks that either focus on general language understanding or specific cybersecurity tasks.

### 1.4.2 Source Data

The dataset includes URLs indicating the sources from which the data was collected. The dataset is available on the following github repository `https://github.com/xashru/cti-bench/tree/main/data`.

### 1.4.3 Intended use

CTIBench is designed to provide a comprehensive evaluation framework for large language models (LLMs) within the domain of cyber threat intelligence (CTI). Dataset designed in CTIBench assess the understanding of CTI standards, threats, detection strategies, mitigation plans, and best practices by LLMs, and evaluates the LLMs' ability to understand, and analyze about cyber threats and vulnerabilities. The intended users of CTIBench are researchers and practitioners in cybersecurity, Cyber threat analysts, incident responders, LLM developers.

## 2 Author statements

### 2.1 Responsibility for Rights Violations

We bear all responsibility in the event of any violations of rights, including but not limited to, intellectual property rights, privacy rights, and any other legal issues that may arise from the use of the dataset described in this submission. We confirm that we have obtained all necessary permissions and consents from data owners and subjects where applicable.

### 2.2 Dataset Licensing

We confirm that the dataset is licensed under the Creative Commons Attribution NonCommercial-ShareAlike 4.0 International.[1] This license allows for the free use, distribution, and modification of the dataset, provided that appropriate credit is given to the original creators, any modifications are indicated, and the use is non-commercial.

## 3 Accessibility

We have hosted our dataset on GitHub at `https://github.com/xashru/cti-bench/tree/main/data` and on Hugging Face at `https://huggingface.co/datasets/AI4Sec/cti-bench`. A permanent DOI identifier is associated with the dataset: DOI: AI4Sec (2024).

We are committed to ensuring the long-term preservation of our dataset through periodic checks to detect and correct any data issues. Additionally, we are dedicated to maintaining this dataset by addressing user queries and issues promptly via email at `ma8235@rit.edu`, and releasing updates and improvements based on user feedback.

## 4 Reproducibility

To ensure reproducibility, we have provided evaluation notebook, response logs from LLMs on our github repository `https://github.com/xashru/cti-bench/tree/main/`.

---

[1] `https://creativecommons.org/licenses/by-nc-sa/4.0/`

## References

[71] AI4Sec. 2024. cti-bench (Revision a86b127). https://doi.org/10.57967/hf/2506