# OpenReview forum: "CTIBench: A Benchmark for Evaluating LLMs in Cyber Threat Intelligence"
_NeurIPS.cc/2024/Datasets_and_Benchmarks_Track — NeurIPS 2024 Track Datasets and Benchmarks Spotlight_

### Official Review · Reviewer_NgSY · 2024-07-24
**An excellent contribution to advance use of LLMs for cyber tasks**

**Rating:** 9
**Confidence:** 5
**Correctness:** Dataset is constructed soundly.  Eval…
**Clarity:** Paper is well written and clear.

**Review:**

This is a valuable contribtion to the very new field of using LLMs for CTI tasks.

It provides a good metric to guide future development of models for CTI tasks.  And a framework which could be added to in the future, with additional tasks or more questions under each task.

I believe this will enable lots of future research.

**Strengths:**

Well written.  Clear purpose.  Well chose and clear measures.  Immediately useful.

**Additional Feedback:**

NA

**Documentation:**

documentation and detail is sufficient.

**Ethics:**

no concerns.

**Limitations:**

Authors addressed limitations sufficiently.

**Opportunities For Improvement:**

No discussion of future work.  Could lay out plans for growing/extending this benchmark with addition questions, more samples for existing questions.

**Relation To Prior Work:**

Paper clearly cites related prior work.

**Summary And Contributions:**

This paper contributes a new benchmark to assess the capability of LLMs across a variety of cyber threat intelligence tasks. The tasks include multiple choice CTI questions gathered from sourced such as MITRE ATT&CK, formatted by GPT, and human validated, root cause mapping, vulnerability severity prediction, and threat actor attribution.  All challenging tasks that require significant reasoning.  The results of assessing current LLMs with these tasks are illuminating.

---

> ### Author Rebuttal · Authors · 2024-08-17
>
> **1. No discussion of future work. Could lay out plans for growing/extending this benchmark with additional questions and more samples for existing questions:** Our future plans include adding more samples to the existing tasks, creating new CTI-specific tasks, and supporting the evaluation of multilingual CTI reports. We will discuss these in the paper.
>
> We have included a new task, CTI-ATE, as detailed above.

---

### Official Review · Reviewer_e7WG · 2024-07-25
**Promising yet incomplete benchmark to assess LLMs on Cyber threat intelligence**

**Rating:** 8
**Confidence:** 3

**Review:**

This submission provides a solid contribution in the field of cybersecurity, introducing a dataset to assess the capabilities of LLMs to conduct task for cyber threat intelligence. The construction of the dataset is sound, and the experimental section gives insights about the performance of current LLMs on the tasks.

Despite these aspects, the scope of the submission is not clear, and the confusion between a dataset of tasks and a benchmark remain. In my opinion, it is missing a software framework and a sound testing protocol to use the dataset as a robust benchmark. The connection with the application of LLM in the context of CTI is also missing and prevents a good understanding of the actual contribution of the submission.

**Strengths:**

- **Broad diversity of tasks**: the benchmark considers for different tasks tailored to assess specific aspects of LLMs;

- **Limited scope of the tasks**: each task is clearly scoped and takes into account the current capabilities of LLMs in terms of interaction;

- **Openness about the limitations of their approach**: the insights from the experiments are well discussed, and the limitations of the current approach in terms of generalisation, clearly mentioned.

**Additional Feedback:**

A comparative evaluation of CTIBench performance by both LLMs and human experts would provide valuable insights into the benchmark's ability to accurately reflect the knowledge and capabilities required for effective CTI tasks.

Investigating the use of LLMs in real-world CTI scenarios by CTI experts for tasks beyond those proposed within CTIBench would also help build a sound and representative benchmark.

UPDATE

After the rebuttal phase, and with the clarifications and improvements made by the authors, I have updated my rating from 6 to 8. Note that this assumes that all proposed changes are indeed implemented in the revised version, which is not available.

**Clarity:**

The paper is well-written and easy to follow.

Minor issues:
---
Page 4, line 125: missing space
Page 6, line 211: LAMA -> LLAMA

**Correctness:**

The content of the submission is correct, and the cybersecurity aspects are well presented. The construction of the tasks is sound and clear.

**Documentation:**

The submission is accompanied with a documentation detailing the main information regarding the dataset.

**Ethics:**

There is no major ethical concerns with the submission. It would be interesting to add a discussion about the malicious use of LLM by malicious actors to exploit CTI knowledge to their advantage.

**Limitations:**

Limitations of the approaches are discussed in a dedicated section and are reasonable.

**Opportunities For Improvement:**

- **Unclear scope**: it seems that the contribution of this paper is a benchmark, but it is actually more a dataset with different tasks compiled, as the software to run the benchmark is missing. A jupyter notebook is provided but implies that outputs have already been generated using LLMs. Elements to fully reproduce the results, or at least run the benchmark in a consistent and reproducible way, are missing.

- **Limited discussion of the impact**: it is not clear how this benchmark will ensure that LLMs can work reliably in a CTI context. Similarly, an assessment of the alignment of CTIBench with the practical skills of cybersecurity professionals specialised in CTI is missing.

- **Potential data leakage bias**: By construction, all data used for the benchmark are public information that could be present in the training set of LLMs, both from original sources and from articles or reports analysing and commenting vulnerabilities. Therefore, it is unclear whether the success of a LLM lies in the fact that the model is able to memorise well a given instance or instead is able to generalise well to unknown case. The authors addressed this issue for the task CTI-RCM by only including descriptions from 2024, but this remains an issue for the other tasks.

**Relation To Prior Work:**

The Related Work section is clear and refers to the most important work connected to the topic of the submission.

**Summary And Contributions:**

This paper introduces CTIBench, a new benchmark designed specifically for evaluating the performance of Large Language Models (LLMs) in the context of Cyber Threat Intelligence (CTI). The benchmark aims to verify that LLMs have the ability to understand, investigate, and analyze cyber threat reports through the assessment of four capabilities: memorisation, understanding, problem-solving, and reasoning. Four tasks are proposed, for which data and instructions are provided: multiple-choice questions on cyber threat intelligence answering; root cause mapping; vulnerability score prediction; and threat actor attribution.

Experiments are conducted with 3 black-box models and 2 open source models. Larger commercial models tend to perform better, even if results vary according to the task. An analysis of each task is provided to get a better understanding of the strength and shortcomings of models.

---

> ### Author Rebuttal · Authors · 2024-08-17
>
> **1. It seems that the contribution of this paper is a benchmark, but it is actually more a dataset with different tasks compiled as the software to run the benchmark is missing:** CTIBench is a benchmark consisting of several tasks that one can utilize to evaluate the reliability of existing or future LLMs in the CTI domain. We have uploaded code to obtain the response from LLMs for the proposed tasks in our GitHub repository.
>
> **2. Limited discussion of the impact: it is not clear how this benchmark will ensure that LLMs can work reliably in a CTI context:** This benchmark proposes four different tasks that evaluate LLM in knowledge-intensive tasks of CTI: memorization (ability to recall previously learned information), understanding (ability to comprehend the content and context), problem-solving (ability to apply knowledge to address specific challenges), and reasoning (ability to draw logical conclusions and make informed decisions).
> While CTI-MCQ only evaluates the memorization ability of LLMs, CTI-RCM (identifying the root cause of a vulnerability given a vulnerability description), CTI-VSP (predicting the vulnerability severity score given a vulnerability description), and CTI-TAA (identifying threat actor given an attack pattern and techniques) is designed to evaluate whether LLMs can reliably work in CTI context by utilizing CTI specific knowledge, understanding, and reasoning. The evaluation of LLMs with these tasks hence can provide a quantitative measure of how LLMs fare in the CTI context, which is missing in the existing benchmarks.
>
> However, we agree that CTI is a multifaceted domain involving several tasks carried out by security professionals. We selected a few tasks that can evaluate the knowledge acquired by LLMs for the CTI context, but we plan to add more tasks in the future. We discuss a new task (CTI-ATE) above.
>
> **3. An assessment of the alignment of CTIBench with the practical skills of cybersecurity professionals specialized in CTI is missing:** As discussed in the paper, CTI-RCM, CTI-VSP, and CTI-TAA reflect critical tasks in CTI where a cybersecurity professional analyzes vulnerability description, and threat reports to identify weakness, severity score, and threat actor respectively. These tasks require nuanced analysis of the information and understanding of the CTI-specific components and language. While we have discussed these in detail in Sections 3.2, 3.3, and 3.4, we are committed to providing more examples if deemed necessary.
>
> **4. Potential data leakage bias:** We utilize CTI data from 2024 for both CTI-RCM and CTI-VSP. These tasks are designed to evaluate whether LLMs can comprehend the content and context of the threat description and apply knowledge and reasoning to address the specific challenge. Hence, we collected CTI data after the cut-off training data of LLMs.
>
> We evaluate the LLMs on the CTI-TAA for the dataset before and after the LLM knowledge cut-off time. Results are shown in the table below. Except for LLAMA3-8B, other LLMs performed better on datasets available to them during the training, suggesting that memorization can help to a certain extent in this task. We include a similar analysis for the new proposed CTI-ATE task as well.
>
> | Model | Overall(Correct) | Overall(Plausible) | Before(Correct) | Before(Plausible) | After(Correct) | After(Plausible) |
> |--|:--:|:--:|:--:|:--:|:--:|:--:|
> | ChatGPT-4   | 52  |  86  | 58.06|90.32  | 42.10|78.95 |
> | ChatGPT-3.5 | 44 |  62  | 50.0 |  75.0 |43.48|60.87 |
> | Gemini-1.5  | 38 | 74 |     34.61 | 76.92| 41.66 |70.83|
> | LLAMA3-70B  | 52  | 80 | 58.06|83.87 |42.10| 73.68 |
> | LLAMA3-8B   |  28 | 36 |25.0 | 25.0 |28.94 |39.47|
>
> **5. On ethics: There are no major ethical concerns with the submission. It would be interesting to add a discussion about the malicious use of LLM by malicious actors to exploit CTI knowledge to their advantage.** As rightfully noted by the reviewer, there is indeed a risk of malicious use of large language models (LLMs) to exploit CTI knowledge to their advantage. For example: LLMs, if misused, could create and disseminate highly convincing but false threat intelligence reports. These reports might mislead decision-makers, cause misguided resource allocation, or prompt inappropriate responses. The ability of LLMs to generate realistic yet fake content could also erode trust in authentic threat intelligence. Hence, there is an ethical responsibility not only to the LLMs creators and developers but also those who deploy these models.  We will discuss the potential misuse of LLMs in the revised paper.
>
> **6. A comparative evaluation of CTIBench performance by both LLMs and human experts would provide valuable insights into the benchmark's ability to accurately reflect the knowledge and capabilities required for effective CTI tasks:** We agree that a comparative analysis between LLMs and human experts would provide valuable insights into the benchmark's effectiveness in assessing the knowledge and capabilities necessary for CTI tasks. However, due to the resource-intensive nature of conducting a rigorous human evaluation, which involves recruiting CTI experts, it was not feasible within the current submission timeline. Our immediate focus was on validating CTIBench as a benchmark for LLMs. That said, we fully recognize the value of human evaluation and plan to incorporate it in future work.
>
> **7. Investigating the use of LLMs in real-world CTI scenarios by CTI experts for tasks beyond those proposed within CTIBench would also help build a sound and representative benchmark:** CTI is a complex field with numerous threat intelligence activities, and we recognize the potential to expand CTIBench with additional tasks. To address this, we developed a new practical CTI task called CTI-ATE (attack technique extraction) and evaluated LLMs on this task. We plan to explore and incorporate additional tasks in future work.

---

> > ### Comment · Reviewer_e7WG · 2024-08-28
> >
> > I would like to thank the authors for their detailed answers. I appreciate the nice additions proposed by the authors, even if unless I'm mistaken, the revised version is not available, therefore I cannot assess how the changes have been implemented. However, based on the response, I will update my review to reflect the improvement in terms of quality of the submission.

---

### Official Review · Reviewer_svyM · 2024-08-05
**This submission is complete and well-organized for its solved problems, however, it requires some improvements.**

**Rating:** 6
**Confidence:** 3
**Clarity:** Yes.

**Review:**

Pros:
- The study provides a thorough evaluation of multiple state-of-the-art LLMs across various CTI tasks. The analysis of errors and performance trends is well-executed and provides valuable insights.
- The paper is well-organized with clear sections and headings. The tasks and datasets are detailed, making the study easy to follow.
- CTIBench is a new benchmark designed explicitly for CTI, addressing a gap in existing benchmarks. The inclusion of multiple tasks covering different aspects of CTI demonstrates originality.
- The benchmark addresses practical CTI tasks, making it highly relevant for real-world applications.


Cons:
- The benchmark covers a limited subset of CTI tasks, which may not fully represent the domain's complexity.
- Using technical terms without adequate explanation may be challenging for non-experts.
- More comparisons with existing benchmarks could have strengthened the claim of originality.

**Strengths:**

- The benchmark tasks are well-defined and cover a range of essential CTI activities, offering a comprehensive evaluation framework for LLMs in this domain.
- Provides a valuable tool for researchers and practitioners in cybersecurity, facilitating the assessment and improvement of LLMs in practical CTI applications.
- Thorough evaluation of multiple state-of-the-art LLMs across diverse CTI tasks, offering comprehensive insights into their capabilities and limitations.
- Addresses the potential risks of LLMs producing false or unreliable intelligence, emphasizing the importance of accurate and reliable AI in cybersecurity.

**Additional Feedback:**

No.

**Correctness:**

- The claims are correct.
- The dataset is constructed soundly.
- The evaluation methods and experiment design are appropriate and performed correctly.

**Documentation:**

Yes.

**Ethics:**

No.

**Limitations:**

- The author admitted in the limitations section that his benchmark did not consider testing whether the LLM included relevant knowledge in the training data. However, the author can still judge; for example, you can directly input the CVE number in the benchmark data set to each LLM and ask for it. If the LLM can directly answer the detailed information of the CVE, it means that the CVE is in the training data of the LLM, and this type of question can be filtered out.
- CTI-MCQ only considers single-choice questions, and the author removed all the multiple-choice questions with multiple answers. It might be more convincing if multiple-choice questions were included separately.
- In the CTI-MCQ experiment, you can consider letting LLM output the reason before outputting the answer, which may perform better.
- Furthermore, in real-world scenes, it would generate solutions rather than make choices. It is better to make CTI-MCQ fill-in-the-blank questions instead of multiple-choice questions

**Opportunities For Improvement:**

- The tasks selected may not fully represent the real-world applications and challenges faced by CTI professionals. For instance, real-world workers may not have options to choose from.
- The evaluation may not fully represent the global nature of cyber threats, which often involve multilingual and diverse contexts.
- The research relies heavily on data from 2024, which might not reflect the current state of cyber threats and vulnerabilities. The evaluation metrics used, such as accuracy and MAD, might not capture the full performance spectrum of LLMs in practical CTI tasks.
- The study does not fully explore the potential misuse of LLMs to generate false or misleading threat intelligence, which could have serious ethical and social ramifications.

**Relation To Prior Work:**

Yes.

**Summary And Contributions:**

The submission introduces CTIBench, a comprehensive benchmark designed to evaluate the performance of Large Language Models (LLMs) in Cyber Threat Intelligence (CTI). CTIBench includes a variety of tasks, such as multiple-choice questions, root cause mapping, vulnerability severity prediction, and threat actor attribution to assess LLMs' understanding, reasoning, and problem-solving capabilities in CTI.

The primary contributions of this submission are the creation of the CTIBench benchmark, the development of diverse CTI-specific tasks, and the evaluation of state-of-the-art LLMs on these tasks. CTIBench addresses the gap in evaluating LLMs' practical applicability in cybersecurity, providing a robust tool for the research community to enhance incident response and threat mitigation.

---

> ### Author Rebuttal · Authors · 2024-08-17
>
> **1. The benchmark covers a limited subset of CTI tasks:** In this work, we narrowed down the most critical tasks for security analysts in their daily jobs that could serve as a benchmark for evaluating the reliability of large language models in CTI. However, we agree that CTI is a multifaceted domain involving several different tasks, including but not limited to vulnerability, weakness, advanced persistent threats, and attack patterns. We have discussed this in the limitation section 7. As discussed above, we have also created a new task: CTI-ATE.
>
> **2. Using technical terms without adequate explanation:** We will include a glossary of terms with their definitions in the appendix.
>
> **3. More comparisons with existing benchmarks:** We have included representative works in general LLM evaluation and security-specific evaluation of LLMs in Section 2 and discussed the differences with our work. Below, we summarize the differences between CTIBench and other cybersecurity benchmarks.
> | Benchmark | Dataset | Goal |
> |-|-|-|
> | SecQA [1] |MCQ|Assess knowledge of computer system security based on a textbook|
> | SecEval [2] |MCQ|Evaluate cybersecurity knowledge|
> | CyberSecEval [3] |Insecure coding | Evaluate LLMs' propensity to generate insecure code|
> ||Cyber attack| Ask LLM to help carry out cyber-attacks|
> | SevenLLM [4] | Understanding | Identify entity, relation, event, malware names, protocol, or algorithm from the given security description |
> | | Generation| Analyze the report and generate summaries by identifying means and methods of attacks |
> | CTIBench | CTI-MCQ| Measure LLM knowledge of CTI domain |
> || CTI-VSP | Compute vulnerability scores|
> | | CTI-RCM| Identify the root cause of vulnerability|
> | | CTI-TAA  | Identify threat actors given a threat intel|
>
> **4. The evaluation may not fully represent the global nature of cyber threats:** We agree with the reviewer on the multilingual nature of cyber threats. We have discussed this as our limitation in Section 7. Our future work will incorporate multilingual CTI evaluations for LLMs.
>
> **5. The research relies heavily on data from 2024, which might not reflect the current state of cyber threats and vulnerabilities:** We utilize CTI data from 2024 for the CTI-RCM and CTI-VSP tasks. These tasks are designed to evaluate whether LLMs can comprehend the content of the threat description and apply knowledge and reasoning to address the specific challenge. Because all the LLMs we evaluated had training data cut-off before 2024, this ensures that the LLMs are not directly recalling data memorized during training. For other tasks, we used data beyond 2024. We use the evaluation metrics accuracy and MAD, which are appropriate for the task.
>
> **6. The study does not fully explore the potential misuse of LLMs to generate false or misleading threat intelligence:** We appreciate the reviewer’s concern about the potential misuse of LLMs for generating false or misleading threat intelligence, which raises significant ethical and social issues. However, we would like to clarify that our study focuses specifically on benchmarking the performance of LLMs across a set of defined tasks. Our work aims to evaluate these models' technical capabilities rather than explore their broader ethical implications, which are important but beyond the scope of this paper.
>
> **7. The author admitted in the limitations section that his benchmark did not consider testing whether the LLM included relevant knowledge in the training data. :** We believe there is some misunderstanding here. We have not discussed “testing LLM includes relevant knowledge” in the limitation section. We carefully created tasks CTI-RCM and CTI-VSP by collecting data beyond the training cut-off of LLMs so that the evaluation accurately represents the understanding ability of LLMs given the security details they have not encountered in the training data.
>
> **8. In the CTI-MCQ experiment, you can consider letting LLM output the reason before outputting the answer:**
> We re-ran the LLM evaluation on MCQ by changing the prompt to one where we asked the LLM to reason before outputting the answer. As the table shows, only GPT3.5 showed significant performance improvement; for LLAMA3-8B, it hurt performance. The brackets represent the approximate number of tokens generated (in thousand tokens).
> | **Model**     | **Acc Without Reasoning** | **Acc With Reasoning**  |
> |---|:---:|:--:|
> | ChatGPT-4 | 71.00 (105) | 71.84 (618) |
> | ChatGPT-3.5 | 54.08 (10)| 59.16 (227)|
> | Gemini-1.5 | 65.44 (5)| 65.68 (360)|
> | LLAMA3-70B | 65.72 (1) | 65.48 (230)|
> | LLAMA3-8B| 61.32 (5) | 55.80 (269)|
>
> **9. CTI-MCQ only considers single-choice questions:** We removed questions that had multiple correct answers from the MCQ dataset to remove ambiguous questions. Our new CTI-ATE task can be considered a multiple-choice question since the model has to extract the set of attack patterns from the given text out of all the possible attack patterns.
>
> **10. The tasks selected may not fully represent real-world applications:** CTI-MCQ is the only task with multiple options. It was designed to evaluate LLMs' CTI knowledge. CTI-RCM, CTI-VSP, and CTI-TAA represent practical CTI tasks where LLMs need to find the cause of vulnerability, predict severity score, and identify threat actors given a threat description. These tasks do not have options, and LLMs need to rely on their knowledge and understanding of the content to provide an answer.
>
> [1] Liu, Zefang. "SecQA: A Concise Question-Answering Dataset for Evaluating Large Language Models in Computer Security." (2023).
> [2] Li, Guancheng, et al. "A comprehensive benchmark for evaluating cybersecurity knowledge of foundation models." (2023).
> [3] Bhatt, Manish, et al. "Purple Llama CYBERSECEVAL: A Secure Coding Benchmark for Language Models." (2023).
> [4] Ji, Hangyuan, et al. "SEvenLLM: Benchmarking, Eliciting, and Enhancing Abilities of Large Language Models in Cyber Threat Intelligence."(2024).

---

### Author Rebuttal · Authors · 2024-08-17

**New Task CTI-ATE (Attack Technique Extraction):**
We introduced an additional task in the benchmark to assess LLMs in a real-world CTI context. The new task, Attack Technique Extraction (ATE), requires identifying the specific attack patterns in the given attack threat behavior and mapping them to the corresponding [MITRE ATT&CK technique IDs](https://attack.mitre.org/techniques/enterprise/). These IDs represent unique techniques that adversaries use during various stages of an attack.

Consider the following [example:](https://attack.mitre.org/software/S0163/) “Janicab is an OS X trojan that exploited a valid Apple Developer ID and unsuspecting users for installation. It captured audio and screenshots, which were then transmitted to a command and control (C2) server. For persistence, Janicab utilized a cron job on Mac devices. The trojan's use of a legitimate Apple Developer ID allowed it to bypass security restrictions by signing the code .”
The description here contains the following attack technique IDs:
T1123 – Audio Capture
T1053 – Scheduled Task
T1113 – Screen Capture
T1553 – Subvert Trust Controls

This task is valuable for CTI practitioners, as accurately identifying the attack techniques used in an incident is crucial for designing effective mitigation strategies and implementing targeted security measures. By mapping specific behaviors to the corresponding MITRE ATT&CK technique IDs, security teams can better understand the adversary's tactics and take proactive steps to counteract them.

**Data collection:** We collected data for this task from the [MITRE ATT&CK framework](https://attack.mitre.org/software/), which offers detailed descriptions of various malware and their adversarial behaviors, each categorized by unique IDs derived from open-source reports. Our dataset includes 30 instances of malware along with their associated attack pattern IDs reported in 2024—information beyond the knowledge cutoff of the LLMs under evaluation. We also incorporated 30 instances of malware reported before 2024. For this task, we focused solely on techniques (excluding sub-techniques). In total, the dataset encompasses 397 attack techniques. The dataset is available in the Hugging Face repository (cti-ate) [here](https://huggingface.co/datasets/AI4Sec/cti-bench).

**Evaluation:** We use the Micro-F1 score as the evaluation metric for this task. ChatGPT-4 outperforms other models, highlighting the task's complexity. Most models perform slightly better on samples collected before their knowledge cutoff date, except for Gemini. However, the difference is not significant, suggesting it can be a good indicator of the LLM's performance on unseen threat reports.

| Model       | Overall (F1) | Before Cutoff (F1) | After Cutoff (F1) |
|-------------|:------------:|:------------------:|:-----------------:|
| ChatGPT-4   |    0.6388    |       0.6542       |      0.6208       |
| ChatGPT-3.5 |    0.3108    |       0.3420       |      0.3333       |
| Gemini-1.5  |    0.4612    |       0.4360       |      0.5263       |
| LLAMA3-70B  |    0.4720    |       0.4934       |      0.4297       |
| LLAMA3-8B   |    0.1562    |       0.1813       |      0.1366       |

---

### Decision · Program_Chairs · 2024-09-26

**Decision:**

Accept (Spotlight)

**Comment:**

This paper proposed a benchmark for cyber threat intelligence. All reviewers recognize the novelty and importance of this benchmark.